# Recombinant Expression in *Pichia pastoris* System of Three Potent Kv1.3 Channel Blockers: Vm24, Anuroctoxin, and Ts6

**DOI:** 10.3390/jof8111215

**Published:** 2022-11-17

**Authors:** Jesús Borrego, Muhammad Umair Naseem, Al Nasar Ahmed Sehgal, Lipsa Rani Panda, Kashmala Shakeel, Attila Gaspar, Cynthia Nagy, Zoltan Varga, Gyorgy Panyi

**Affiliations:** 1Department of Biophysics and Cell Biology, Faculty of Medicine, University of Debrecen, 4032 Debrecen, Hungary; 2Department of Inorganic and Analytical Chemistry, Faculty of Science and Technology, Institute of Chemistry, University of Debrecen, 4032 Debrecen, Hungary

**Keywords:** *Pichia pastoris*, recombinant peptides, scorpion venoms, Kv1.3 blockers

## Abstract

The Kv1.3 channel has become a therapeutic target for the treatment of various diseases. Several Kv1.3 channel blockers have been characterized from scorpion venom; however, extensive studies require amounts of toxin that cannot be readily obtained directly from venoms. The *Pichia pastoris* expression system provides a cost-effective approach to overcoming the limitations of chemical synthesis and *E. coli* recombinant expression. In this work, we developed an efficient system for the production of three potent Kv1.3 channel blockers from different scorpion venoms: Vm24, AnTx, and Ts6. Using the *Pichia* system, these toxins could be obtained in sufficient quantities (Vm24 1.6 mg/L, AnTx 46 mg/L, and Ts6 7.5 mg/L) to characterize their biological activity. A comparison was made between the activity of tagged and untagged recombinant peptides. Tagged Vm24 and untagged AnTx are nearly equivalent to native toxins in blocking Kv1.3 (Kd = 4.4 pM and Kd = 0.72 nM, respectively), whereas untagged Ts6 exhibits a 53-fold increase in Kd (Kd = 29.1 nM) as compared to the native peptide. The approach described here provides a method that can be optimized for toxin production to develop more selective and effective Kv1.3 blockers with therapeutic potential.

## 1. Introduction

Voltage-gated potassium (Kv) channels regulate the selective efflux of K^+^ through the plasma membrane, thereby shaping the action potentials in excitable cells and regulating many other cellular functions in non-excitable cells [1]. The Kv channel family includes 12 subfamilies (Kv1-Kv12). The Kv1 (Shaker) subfamily consists of eight voltage-gated potassium channels belonging to the delayed rectifier class (Kv1.1–Kv1.8) [2]. Of this Kv subfamily, the Kv1.3 channel is mainly expressed in neurons and immune cells [3], and therefore, channel inhibition in these cells may be useful in the therapy for various diseases, such as T-lymphocyte-mediated autoimmune diseases, chronic renal failure, asthma, obesity, type II diabetes mellitus, cognitive disabilities, cardiac arrhythmia, and some cancers [4,5,6,7]. Kv1.3 channel activity can be modulated by metal ions, small organic molecules, and the peptides isolated from venoms [8], which affect Kv channel function by blocking the ion-conducting pore from the inside or outside or by modifying the channel gating by binding to the voltage sensor domain or auxiliary subunits [9]. To date, about 460 toxins are known exclusively for their Kv channels [10]. Some toxins block the Kv1.3 channel in the nanomolar range, e.g., charybdotoxin [11] and noxiustoxin [12], and others can even reach the picomolar range, e.g., Vm24 [13], ShK [14], margatoxin [15], and HsTx1 [16]. The exploration of the therapeutic potential of these toxins, including their extensive investigation and use in preclinical and clinical studies, is mainly hindered by the low yield of purified toxins from milked venom. Therefore, the production of toxin peptides either by chemical synthesis or heterologous expression is the strategy followed to overcome this limitation [17]. The chemical synthesis is complex and not cost-effective for producing mutant toxins for structure–activity relationship studies [18]. It also requires extensive screening of in vitro folding conditions [19]. To date, *Escherichia coli* (*E. coli*) is the best-known heterologous expression system, in which recombinant peptides frequently accumulate in the cytoplasm, leading to misfolding and aggregation in some cases [20]. The solution to these problems is the periplasmic expression or the use of genetically modified *E. coli* strains, which can enhance disulfide bond formation, but with very low yields [21]. Yeast is a heterologous expression system that has important advantages, including the ability to perform multiple eukaryotic post-translational modifications, fast growth, high biomass concentration, the ability to produce a high amount of recombinant protein, both intracellularly and extracellularly, scalable fermentation, and pathogen-free production [22,23]. Sixty-nine percent of the total scorpion venom components recombinantly expressed in bacteria or yeast correspond to neurotoxins with three to four disulfide bridges. Eighty percent of these neurotoxins were expressed in *E. coli* and twenty percent in yeast. In half of the reports, a yield of 1 to 2 mg/L was observed for the *E. coli* expression system, whereas a yield of more than 9 mg/L was obtained in yeast. In addition to this difference in yield, the misfolding of the peptide in *E. coli* had to be overcome in some cases [22]. It should be noted that in our research group, there were cases in which peptide expression succeeded with *Pichia pastoris* (recently reclassified as *Komagataella phaffii*) but without biological activity (unpublished data), which is similar to the results reported for the expression of the I5A toxin using the same recombinant expression system [24]. Therefore, further studies on the expression of these peptides in yeast are needed to improve the comparative analysis of these two expression systems (bacteria and yeast). These results will lead to the improvement in the recombinant production of peptides in sufficient amounts that retain their biological activity in the performance of studies for the development of new drugs, bioinsecticides, antibodies, etc.

In this work, we report the recombinant production of three potent Kv1.3 peptide blockers with four disulfide bridges (anuroctoxin, Ts6, and Vm24) using the yeast *Pichia pastoris* as an expression system. Anuroctoxin (AnTx, α-KTx 6.12) is a small peptide consisting of 35 amino acids (4109 Da). It was isolated from the venom of the Mafia scorpion (*Anuroctonus phaiodactylus*), and it exhibits a Kd = 0.73 nM for the block of Kv1.3 [25]. Ts6 (α-KTx 12.1) consists of 40 amino acids with a molecular mass of 4506 Da. Ts6 was isolated from the venom of the scorpion *Tytyus serrulatus* and inhibits hKv1.3 currents with a Kd = 0.55 nM [26]. Vm24 (α-KTx 23.1) is a 36-residue peptide (3864 Da) isolated from the venom of the scorpion *Vaejovis mexicanus smithi* [27]. It has the highest affinity for Kv1.3 among these peptides; it inhibits Kv1.3 with a Kd = 2.9 pM [13]. After recombinant expression of the peptides, they were characterized by mass spectrometry, and the activity was tested by a patch clamp to compare the Kd value of the recombinant peptides with the activity previously reported for the native peptides.

## 2. Materials and Methods

### 2.1. Chemicals

All chemicals, unless stated otherwise, were purchased form Sigma-Aldrich (Budapest, Hungary).

### 2.2. Plasmid Construction

*Pichia pastoris* X-33 (Invitrogen, Waltham, MA, USA, Cat. No. K1740-01) was used as the expression host. The amino acid sequences of AnTx, Ts6, and Vm24 were retrieved from the Uniprot database https://www.uniprot.org/ (accessed on 10 March 2021) (AnTx: P0C166, Ts6: P59936, and Vm24: P0DJ31). The sequences were reverse translated, and the gene cassettes were designed by adding the coding sequence of the 6x-HisTag at the N-terminus to facilitate the protein purification by Ni-NTA affinity chromatography. In addition, a factor Xa protease site (IEGR) was added in the case of Vm24, and an enterokinase protease site (DDDDK) was added in the case of AnTx and Ts6 to remove the His-tag of the expressed toxin if needed. The gene cassettes were codon-optimized for *P. pastoris* according to the codon usage database available at www.kazusa.or.jp/codon (accessed on 10 March 2021) and synthesized by Integrated DNA Technologies, Belgium. The designed gene sequences were cloned into the yeast expression vector pPICZαA (Invitrogen, Waltham, MA, United States) using *PstI* and *SalI* for AnTx and Vm24, and *KpnI* and *EcoRI* for Ts6. The in-frame ligation and nucleotide sequence of AnTx, Ts6, and Vm24 were confirmed by DNA sequencing.

### 2.3. Pichia Trasnformation

To generate the toxin-expressing strains, each expression plasmid was linearized by digestion with the restriction enzyme *SacI* at 37 °C overnight and transformed into *P. pastoris* X-33 competent cells using the Pichia EasyComp Transformation Kit (Invitrogen, United States, Cat. No. K173001) according to the protocol indicated by the manufacturer. The transformed X-33 cells were spread on a YPD agar plate (1% yeast extract, 2% peptone, 2% dextrose, 2% agar, pH 7.0) containing 200 µg/mL Zeocin (Invitrogen, United States, Cat. No. R25005). After 72 h incubation at 30 °C, the 15–25 biggest colonies were grown on YPD plates supplemented with progressively increasing amounts of Zeocin (500, 1000, and 2000 µg/mL) to select the clone that exhibited hyper-resistance to Zeocin. To confirm the integration of the expression construct into the genome of the *Pichia* transformants, a colony PCR using plasmid and gene-specific primers was performed on colonies grown on 2000 µg/mL Zeocin.

### 2.4. Protein Expression and Purification

Three to nine selected clones from the YPD plate containing 2000 µg/mL Zeocin were grown overnight in 5 mL YPD medium (1% yeast extract, 2% peptone, 2% dextrose, pH 7.0) and diluted in 10 mL BMGY medium (1% yeast extract, 2% peptone, 100 mM potassium phosphate, pH 6.0, 1.34% YNB, 4 × 10^−5^% biotin, and 2% glycerol) to an OD_600_ = 0.2. The clones were incubated at 30 °C at 240 rpm for 24 h. The cells were collected by centrifugation, resuspended in 2 mL of BMMY induction medium (the same as BMGY with 1% methanol instead of glycerol), and grown for 72 h at 30 °C with constant shaking (240 rpm). Absolute methanol was added every 24 h at a final concentration of 1% to maintain protein induction. For each sample, 1 mL of supernatant was collected, and protein was precipitated with the trichloroacetic acid protocol and analyzed on 16% tricine SDS-PAGE [28] (in the case of AnTx, 30 µL of supernatant was loaded directly onto the gel). The clone with the highest expression level of each toxin was selected and cultured in 100 mL BMGY (30 °C at 240 rpm for 24 h); then, the cells were collected by centrifugation and resuspended in 50 mL BMMY. Protein production was induced for 72 h at 30 °C with constant shaking (240 rpm). The cultured supernatant was separated by high-speed centrifugation and diluted to two-fold in potassium phosphate buffer, and imidazole was added at a final concentration of 30 mM. The filtered supernatant was loaded on a Ni SepharoseTM 6 Fast Flow (Cytiva, Uppsala, Sweden AB, Cat. No. 17531801) manually packed column pre-equilibrated with a binding buffer (25 mM potassium phosphate, 300 mM NaCl, 30 mM imidazole, pH 7.4) at a flow rate of 1 mL/min using a peristaltic pump (miniplus 3, GILSON^®^). The column was washed with three column volumes (CVs) of wash buffer (25 mM potassium phosphate, 300 mM NaCl, 40 mM imidazole, pH 7.4), and the peptides were eluted by running three CVs of elution buffer (25 mM potassium phosphate, 400 mM imidazole, pH 7). The fractions collected from the affinity column were further purified by reverse-phase high-performance liquid chromatography (RP-HPLC) using a C18 analytical grade reverse-phase column (4.6 mm × 250 mm, 5 μM bead size, Vydac^®^ 218TP) using a Prominence HPLC system (Shimadzu, Germany) at a flow rate of 1 mL/min. The protocol for each toxin was performed as follows. Recombinant anuroctoxin (AnTx) was purified using a linear gradient of 0–25% of solvent B (0.1% TFA in 100% CH_3_CN) in solvent A (0.1% TFA in water) over 35 min. Recombinant Vm24 and Ts6 were purified using a linear gradient of 10–40% of solvent B in solvent A over 60 min. To eliminate the salts, 100% solvent A was flowed (5 min for Vm24 and Ts6, and 3 min for AnTx) before the gradient protocol. The absorbance at 230 nm was monitored with a PDA detector. The peak fractions were collected manually and tested on 16% tricine SDS-PAGE. After toxin characterization (mass spectrometry and electrophysiology), protein production was performed on a large scale to obtain a sufficient peptide amount for further experiments. The protocol used was the same as above with the following exceptions. The clone was inoculated in YPD medium containing 200 µg/mL Zeocin and grown overnight. Three 2 L flasks containing 500 mL BMGY medium were inoculated with the cells from the YPD medium. Twenty-four hours later, the cells were centrifuged and resuspended in 330 mL BMMY to obtain 1 L of culture.

### 2.5. Cleavage of the His-Tag

The 6x-His-tag fused at the N-terminus of AnTx and Ts6 was cleaved with enterokinase light chain (New England, Biolabs, Ipswich, MA, USA, Cat. No. P8070L). The reaction mixture was prepared as follows. Twenty-five micrograms of peptide was mixed with 1 μL enterokinase at an enzyme-to-substrate ratio of 1:100 in TBS buffer (20 mM Tris-HCl, 50 mM NaCl, 2 mM CaCl_2_, pH 8.0) and incubated overnight at 25 °C. To separate the untagged peptide from the tagged one, the reaction mixture was passed through a Ni Sepharose manually packed column. The eluate was purified by RP-HPLC using the protocols described under the heading of protein expression and purification.

### 2.6. Mass Spectrometry Analyses

Mass spectrometric determinations were performed using an ESI QTOF-MS instrument (maXis II UHR ESI-QTOF MS, Bruker, Bremen, Germany). The mass spectrometer was operated in positive ionization mode; 0.5 bar nebulizer pressure, 200 °C dry gas temperature, 4 L/min dry gas flow rate, 4000 V capillary voltage, 500 V end plate offset, 1 Hz spectra rate, and 500–2500 *m*/*z* mass range were applied. ESI tuning mix (Agilent) calibrant injected after each run enabled internal *m*/*z* calibration. Mass spectra were processed and evaluated by Compass Data Analysis version 4.4 (Bruker).

### 2.7. Cells

Human venous blood from anonymized healthy donors was obtained from a blood bank. Peripheral blood mononuclear cells (PBMCs) were isolated by density gradient centrifugation using Histopaque1077 (Cat. No 10771). The obtained cells were resuspended in RPMI 1640 medium containing 10% fetal calf serum, 100 μg/mL penicillin, 100 μg/mL streptomycin, and 2 mL glutamine, seeded at a density of 5 × 10^5^ cells per ml in a 24-well culture plate and grown in a 5% CO_2_ incubator at 37 °C for 2–5 days. Phytohemagglutinin A (Cat. No. L1668) was added to the medium at concentrations of 5, 7, and 10 μg/mL to increase the voltage-gated potassium ion channel expression.

### 2.8. Electrophysiology

Electrophysiological measurements were performed by using the patch clamp technique in a whole-cell configuration on voltage-clamped cells using the Axon Multiclamp 700B amplifier and the Axon Digidata1440 digitizer; the Clampex 10.7 software was used for data acquisition (all from Molecular Devices, Sunnyvale, CA, USA). Micropipettes were pulled from GC150F-15 borosilicate capillaries (Harvard Apparatus Kent, Edenbridge, UK), resulting in 3–4 MΩ resistance in the bath solution. The extracellular solution (bath solution) contained 145 mM NaCl, 5 mM KCl, 1 mM MgCl_2_, 2.5 mM CaCl_2_, 5.5 mM glucose, and 10 mM HEPES, with a pH of 7.35 and an osmolarity between 302 and 308 mOsM/L. The pipette filling solution (intracellular) consisted of 140 mM KF, 2 mM MgCl_2_, 1 mM CaCl_2_, 10 mM HEPES, and 11 mM EGTA, with a pH of 7.22 and an osmolarity of 295 mOsM/L. When the toxins were dissolved in bath solution at different molar concentrations, they were supplemented with 0.1 mg/mL bovine serum albumin (BSA). To record hKv1.3 in activated human T lymphocytes (Kv1.3 overexpression), 15 ms-long depolarizing pulses were applied at +50 mV from a holding potential of −120 mV every 15 s. Current traces were low-pass filtered by the analog four-pole Bessel filters of the amplifiers, and the sampling frequency was set at 20 kHz, which was at least twice that of the filter cutoff frequency. The control and test solutions were perfused into the cell through a gravity flow perfusion system. The excess bath solution was constantly removed with vacuum suction.

### 2.9. Patch Clamp Data Analyses

The effect of the toxin at a given molar concentration was calculated as the remaining current fraction (RCF = I/I_0_, where I_0_ is the peak current in the absence of the toxin and I is the peak current at the equilibrium block at a given toxin concentration). The data points on the dose-response curves represent the mean of 3–5 individual measurements. The data points were fitted with the Hill equation,
$$\mathrm{RCF}=\frac{K_{d}^{H}}{K_{d}^{H}+{\left[\mathrm{toxin}\right]}^{H}}$$
where [toxin] is the concentration of the toxin and H is the Hill coefficient. The best fit curve gave the K_d_ value of a given toxin.

### 2.10. Bioinformatics

The structural visualization was conducted with the software PyMol v2.3.2. Vm24 (2K9O), and the Ts6 (1C56) structures were retrieved from the Protein Data Bank (PDB) https://www.rcsb.org/ (accessed on 17 August 2022).

### 2.11. Statistics

Data are expressed as means ± SEM. Statistical analyses and graph plotting were executed in GraphPad Prism software (version 8.0.1) (San Diego, CA, USA).

## 3. Results and Discussion

In this work, the recombinant expression of three peptides from scorpion venom in the *P. pastoris* expression system is reported (AnTx, Vm24, and Ts6). Previously, our group attempted to express the AnTx peptide using *E. coli*. Soluble expression of the peptide was not achieved in the cytoplasm, and periplasmic expression yielded only a very low amount of the toxin. The expression of these disulfide-rich peptides in *E. coli* may result in misfolding, causing peptide aggregation so that the protein is formed in inclusion bodies rather than in soluble form [29,30]. Therefore, the expression of AnTx was tested in *Pichia*, and based on the yield observed, we decided to use the same model for the expression of the other peptides, Vm24 and Ts6.

### 3.1. Plasmid Construction

After digestion of the toxin genes and the empty pPICZαA vector with the restriction enzymes (*PstI* and *SalI* for AnTx and Vm24, and *KpnI* and *EcoRI* for Ts6), the toxin genes were cloned downstream of the *P. pastoris* AOX1 promoter and in-frame with the α-factor of the pPICZαA vector (Figure 1). The resulting recombinant vectors (pPICZαA-AnTx, pPICZαA-Vm24, and pPICZαA-Ts6) were sequenced to verify the nucleotide sequence of the toxin and to confirm that the sequence was inserted in-frame with the α-factor secretion signal. Competent *P. pastoris* X-33 cells were transformed with the linearized plasmid, and it yielded approximately 20 to 30 Zeocin resistance colonies upon 72 h incubation at 30 °C. The largest colonies were selected for testing for hyper-resistance to Zeocin. A colony PCR was performed on the colonies, showing the best growth at 2000 µg/mL Zeocin. The colonies with the best growth that showed an expected PCR amplicon were selected for the protein expression assay.

### 3.2. Peptide Expression and Purification

For each toxin, 3–9 clones were selected to evaluate the protein expression. Figure 2 shows the tricine SDS-PAGE, where the expression of the recombinant peptides was confirmed. The bands of the recombinant peptides were expected at ~6 kDa. For Vm24 (Figure 2a), a band below 6.5 kDa can be seen. However, for AnTx (Figure 2b) and Ts6 (Figure 2c), the band indicates a molecular weight of around 10 kDa. This effect of reduced electrophoretic mobility has been observed in other similar peptides that also have a high proportion of basic amino acids [30,31]. Moreover, as discussed later, the masses of the recombinant peptides determined by mass spectrometry agreed with the expected masses calculated for the fully oxidized peptides.

The selected clones of each toxin were grown in a larger culture (50 mL BMMY). The His-tagged peptides were separated from the supernatant by Ni-NTA affinity chromatography. The obtained samples were then loaded into an RP-HPLC column to purify the recombinant peptides. The HPLC fractions obtained in this purification step were analyzed using SDS-PAGE. Figure 3 shows the HPLC fraction corresponding to each recombinant peptide. These fractions were characterized by mass spectrometry and their biological activity on the Kv1.3 channel was assayed. The experimental molecular mass of the three recombinant peptides was consistent with the expected theoretical molecular mass (Table 1) (Appendix A).

### 3.3. Biological Activity of the Recombinant Peptides on the Kv1.3 Channel

In the preliminary experiments to evaluate the biological activity of the recombinant toxins, peptide concentrations slightly higher than the reported Kd values for the native toxins were used. As can be seen in Figure 4, the concentrations of Vm24 (10 pM) and AnTx (10 nM) blocked more than 50% of the current (reported Kd = 2.9 pM and 0.73 nM, respectively). However, this effect could not be replicated with 100 nM Ts6 toxin (180 times the Kd for the native toxin. To determine whether the recombinant Ts6 was biologically active, a concentration of 300 nM (a concentration well above the reported Kd = 0.55 nM) was tested, which blocked ~22% of the Kv1.3 current. It should be noted that all these measurements were conducted with 6x-His-tagged peptides. Although this tag has some advantages, such as small size, no electrical charge, low toxicity, and immunogenicity, it also has the disadvantage that it can sometimes affect protein structure and function [32,33]. Therefore, the effect of the His-tag on the interaction between the peptides and the channel was evaluated by calculating and comparing the Kd values for the tagged and untagged recombinant toxins.

Larger amounts of recombinant peptides were needed to determine the equilibrium dissociation constant (Kd) values of the toxins for the interaction with the hKv1.3 ion channel. The yeast culture was scaled up to 1 L. The yields after the RP-HPLC purification of the tagged Vm24, Ts6, and AnTx produced in *P. pastoris* were 1.6 mg/L, 7.5 mg/L, and 46 mg/L, respectively. The yield of Ts6 was similar to that of other toxins also expressed in yeast, such as TxVIA (10 mg/L) [34], margatoxin (12 mg/L), and agitoxin-2 (14 mg/L) [31]. The outliers were AnTx with a high yield and Vm24 with a low yield. Vm24, AnTx, and Ts6 were previously either isolated directly from venoms or chemically synthesized; therefore, the yields obtained in this study cannot be compared with other studies using the recombinant expression of these peptides. However, as mentioned earlier, higher yields are usually obtained in the recombinant production of neurotoxins when yeast is used as the expression system. Two clear examples are the toxins LqhIT2 and MgTx, for which the yields in *P. pastoris* were 18- [35,36] and 10-fold [15,37] higher, respectively, than in *E. coli*. The heterologous expression of proteins in insect or mammalian cell cultures is complex and expensive and generally results in minute amounts of recombinant proteins. For example, the yield of Psalmotoxin 1 in a Drosophila S2 cell culture was reported to be 0.48 mg/L [38]. On the other hand, cell-free protein synthesis (CFPS) could result in yields ranging from µg to mg per ml in general [39]. For example, Ramm et al. reported that the expression of AB_5_ toxins (cholera toxin, heat-labile enterotoxin) in a CFPS system varies between 7 and 20 µg/mL in mammalian CHO cell lysate or insect-based *Sf*21 lysate [40]. Moreover, CFPS is an expensive approach, which is usually exploited to circumvent the limitations of cell-based systems [41,42]. It should be noted that repetition and optimization of the process (1 L culture) were not performed in this work because the amounts of toxins produced were sufficient for the evaluation of the biological activity. However, it is well known that the optimization of the protocols would lead to an increase in the production of this kind of toxins by *P. pastoris* as this has already been observed in the production of margatoxin, where the optimization of the protocol increased the yield threefold (36 mg/L) compared to the previously reported yield [37]. The His-tag of AnTx and Ts6 was removed using enterokinase. After overnight digestion, the untagged peptides were purified using RP-HPLC with yields of untagged peptides ranging from 60 to 75%. The fractions were analyzed by tricine SDS-PAGE (Appendix A) and mass spectrometry (Appendix A). The monoisotopic experimental masses of the HPLC-purified peptides agreed well with the expected masses for the untagged peptides (AnTx: predicted = 4097.806 Da, experimental = 4097.7831 Da; Ts6: predicted = 4502.879 Da, experimental = 4502.8256 Da).

The onset and recovery from the block of hKv1.3 currents at various concentrations of tagged or untagged recombinant toxins are shown in Figure 5. Normalized peak currents (I_norm_ = It/I_0_, where It is the peak current in the presence of the toxin at time t, and I_0_ is the peak current in the absence of the toxin) were plotted as a function of time. In the case of Tag-Vm24, the association kinetics were very slow; ~25 min was needed to achieve an equilibrium block at a 10 pM concentration (Figure 5a), whereas ~5 min (Figure 5b) was needed to reach equilibrium at the 1 nM concentration. The block was not reversible upon perfusing the cell with the Tag-Vm24-free solution (Figure 5a,b). These data show that the binding kinetics of Tag-Vm24 were similar to those reported for native and synthetic Vm24 [13,27]. In the case of AnTx, the association and dissociation rates were moderate; the application of 1 nM of Tag-AnTx or Untag-AnTx developed the steady-state block in a similar time course within ~2.5 min, and the full recovery of the peak current was attained after perfusing the cell with toxin-free solution (Figure 5c,d). However, in the case of Ts6, the association and dissociation kinetics were very fast; the onset of the equilibrium block at progressively increasing concentrations of either Tag-Ts6 (Figure 5e) or UnTag-Ts6 (Figure 5f) and complete recovery of peak current happened in a few seconds. Moreover, the apparent association rate was becoming faster with the increasing concentration of Ts6 (either tagged or untagged), which is also in agreement with the assumption of a simple bimolecular reaction between the toxin and the channel, as described for classical Kv1.3 pore blocker toxins [43]. As Tag-Vm24 is an irreversible blocker, a cumulative concentration–response was obtained for this peptide. In contrast, because of the faster kinetics and reversible blocking effect of AnTx and Ts6, the concentration–response curve was determined by repeatedly perfusing the same cell with the control solution or containing different concentrations of toxin.

The Kd values for Tag-Ts6, Tag-AnTx, and Tag-Vm24 were 886.9 ± 27 nM, 1.29 ± 0.14 nM, and 4.4 ± 0.6 pM, respectively, as determined by fitting the Hill equation to the concentration–response relationships (Figure 6). Tag-Ts6 showed a 1600-fold decrease in potency on the hKv1.3 channel compared with the Kd value reported for native Ts6 (0.55 nM). This was the highest difference between the tagged and the native versions of the three peptides. For the other two peptides, the Kd values were slightly different; Tag-AnTx showed a 1.8-fold increase (native Kd = 0.72 nM), whereas Tag-Vm24 showed a 1.5-fold increase (native Kd = 2.9 pM) in the Kd.

Gurrola et al. reported that there are significant differences between native Vm24 and synthetic Vm24 in the blocking effect on the hKv1.3 channel at concentrations below 3 pM but not at 10 pM or higher [27]. This effect is due to the extremely slow blocking kinetics of Vm24 at low picomolar concentrations, where equilibrium is reached only after 0.5 to 1 h of toxin application, which may lead to an overestimation of the Kd value [13]. For this reason, and because Tag-Vm24 showed biological activity at picomolar concentrations, as well as the fact that the Kd value determined from the fit to the data points was very close to the previously reported value, we decided not to proceed with the evaluation of Untag-Vm24. The Kd values for Untag-AnTx and Untag-Ts6 were 0.72 ± 0.06 nM and 29.1 ± 2.2 nM, respectively. Untag-AnTx showed a slight, 1.8-fold decrease in Kd value compared with Tag-AnTx. This means that His-tag practically did not affect the interaction of recombinant AnTx with the hKv1.3 channel. Even more interesting is that the Kd values of Untag-AnTx and the native AnTx were essentially the same (0.72 nM and 0.73 nM, respectively). Native AnTx has a pyroglutamic acid at the N-terminus region and an amidated C-terminal residue [25]. It has been reported that several scorpion toxins require C-terminal amidation for full biological activity, without which their potency is severely limited. This has been tested either by comparing two related native toxins (with or without amidation) [44] or by comparing the activity of recombinant non-amidated peptide with its native amidated version [45]. It seems that C-terminal amidation could be involved in molecular recognition processes or could reduce the negative charge of the carboxyl group and the peptide as a whole [46]. However, as shown in this work, the interaction of recombinant AnTx with the hKv1.3 channel does not require this posttranslational modification, which is consistent with the similar results reported when comparing the effect of native maurotxin [47] and its synthetic version [48].

On the other hand, Untag-Ts6 showed a drastic change in Kd value when the His-tag was removed (Kd = 29.1 nM). The Kd value of Untag-Ts6 was 30-fold lower than that calculated for Tag-Ts6 (Kd = 886.9 nM). This implies that the additional amino acids at the N-terminus of Tag-Ts6 drastically affect its biological activity compared with the other two peptides (Tag-Vm24 and Tag-AnTx). The amino acid sequences of the three peptides and the structures of Ts6 and Vm24 (the only structures reported so far) were compared to reveal the potential connections between the effects of the peptides on Kv1.3 and the presence of the His-tag. Figure 7a shows the alignment of the Vm24, Ts6, and AnTx amino acid sequences. As shown in Figure 7a, the three peptides exhibit well-conserved disulfide bridges as well as the functional dyad characteristic of this type of toxin. The dyad consists of two highly conserved amino acid residues, lysine and a neighboring aromatic or aliphatic residue, in this case tyrosine or phenylalanine. The β-sheet side of the toxin faces the entrance of the channel pore, and the lysine side chain faces the selectivity filter, in this way occluding the pore [49]. Vm24 and AnTx have a fourth disulfide bridge conserved between them. The structural studies of AnTx show a three-dimensional structure virtually identical to that of Vm24, which may explain why the His-tag had the same effect in both recombinant peptides [25,50]. Figure 7b shows the structural alignment between Vm24 and Ts6, in which the functional dyad in both toxins is oriented in the same direction, indicating that the mechanism of interaction with the channel should be very similar. However, as shown in Figure 7c, the N-terminus of Ts6 has two differences from the N-terminus of Vm24: it is longer and has a disulfide bridge. Therefore, the length of the N-terminus, together with the disulfide bridge that could make it less flexible, may result in the His-tag physically hindering the entry of the toxin to its receptor site and the subsequent interaction of the dyad Lys (Lys30) with the channel pore. The functional dyad of AnTx, Vm24, and Ts6 is located (as in many other related toxins) at the C-terminus, which is the region that plays an important role in the interaction with the channel and thus leads to the blocking of the channel. As observed in this work, although the His-tag was expressed at the N-terminus of the toxins, it had a very different effect depending on which toxin it was conjugated to. There are also other types of affinity tags that can be used in the expression of peptides, e.g., GST, MBP, FLAG, etc. [33], which may or may not affect their biological activity. Finding a pattern between the features of the toxin and the features of the desired affinity tag could be useful in obtaining a conjugated toxin where the tag has little or no effect on biological activity (as in the case of Tag-Vm24 and Tag-AnTx), thus eliminating the need for tag removal and the subsequent purification steps.

As mentioned earlier, the Kd values of Tag-Vm24 and Untag-AnTx were very similar to those reported for the native toxins. However, the Kd value of Untag-Ts6 was increased 53-fold compared with its native version (Kd = 0.55 nM). The Kd value of the native Ts6 was determined using Xenopus oocytes expressing rat Kv1.3 (rKv1.3). In this work, recombinant Vm24, AnTx, and Ts6 were tested on hKv1.3 expressed in activated PBMCs (the identical model used for reporting the data for the native Vm24 and AnTx toxins). Differences in the type of cell expression system as well as the use of a channel from different species could be the cause of the discrepancy in the results. This phenomenon has been also reported for maurotoxin. While the Kd value for rKv1.3 expressed in the oocytes was 180 nM [37], the hKv1.3 expressed in the CHO cells was almost insensitive to maurotoxin at a concentration of 100 nM [51]. Table 2 summarizes the Kd values calculated for the recombinant peptides.

Our results support the proposal that *Pichia pastoris* is a reliable heterologous expression system for the expression of toxins from scorpion venoms. It is worth noting that the expression protocols can still be optimized to increase the yield, especially in the case of Vm24. However, as described here, *Pichia* expresses the toxins Vm24, AnTx, and Ts6, which inhibit the hKv1.3 channel in a similar manner to their native versions, thereby confirming the power of *Pichia* as a recombinant expression system for cysteine-rich peptides.

## Figures and Tables

**Figure 1 jof-08-01215-f001:**
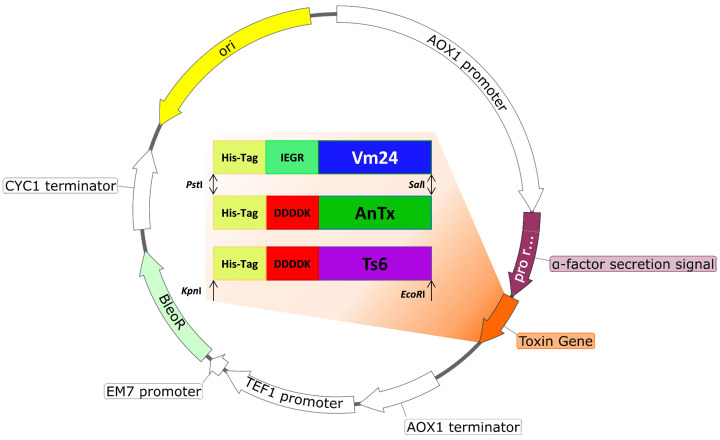
Graphical representation of recombinant plasmid pPICZαA-Vm24, -AnTx, and -Ts6.

**Figure 2 jof-08-01215-f002:**
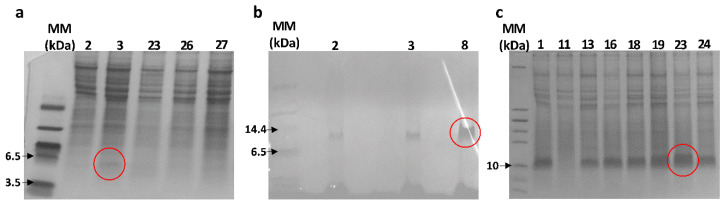
Tricine SDS-PAGE analysis of the different clones expressing the recombinant peptides. MM: molecular marker (kDa). The figure shows the 16% tricine SDS-PAGE analysis of the expression of recombinant Vm24 (**a**), AnTx (**b**), and Ts6 (**c**) by the clones growing at 2000 µg/mL zeocin (numbers above the lines refer to the clone numbers). The red circle indicates the selected clones that showed the highest protein expression.

**Figure 3 jof-08-01215-f003:**
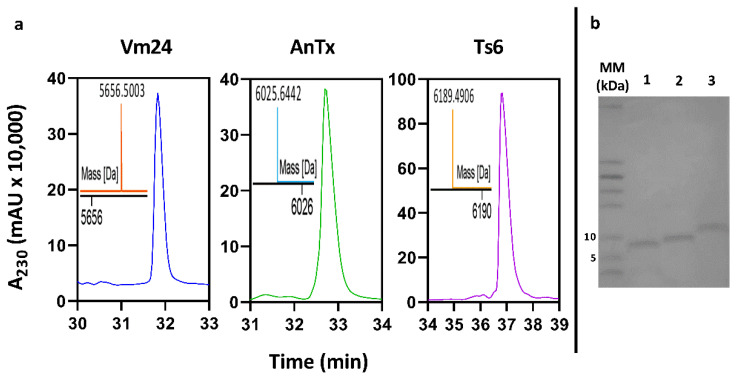
RP-HPLC purification of Vm24, AnTx, and Ts6. (**a**) RP-HPLC chromatograms of the recombinant peptides. Chromatograms show the fractions that are biologically active on the hKv1.3 channel. Acetonitrile % of elution and retention time for recombinant peptides were: Vm24 23.4%/31.8 min, AnTx 21.2%/32.7 min, and Ts6 25.9%/36.8 min. The inset in each chromatogram represents an ESI-QTOF-MS spectrum showing the monoisotopic mass of the respective fraction. (**b**) Sixteen percent tricine SDS-PAGE of the purified peptides. MM: molecular marker (kDa), line 1: Vm24, line 2: Ts6, and line 3: AnTx.

**Figure 4 jof-08-01215-f004:**
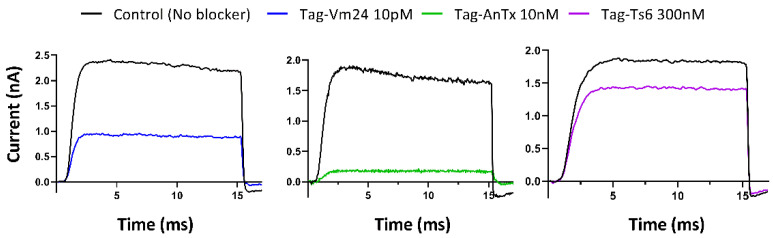
Inhibition of hKv1.3 currents by tagged recombinant Vm24, AnTx, and Ts6. These peptides are indicated in the figure as Tag-Vm24, Tag-AnTx, Tag-Ts6. Whole-cell currents through hKv1.3 were evoked from activated human peripheral T lymphocytes by depolarization to +50 mV from a holding potential −120 mV for 15 ms duration. Test pulses were applied every 15 s. Representative traces show the K^+^ current before the application of toxin (control) and after reaching the equilibrium block in the presence of Tag-Vm24 (blue), Tag-AnTx (green), and Tag-Ts6 (purple).

**Figure 5 jof-08-01215-f005:**
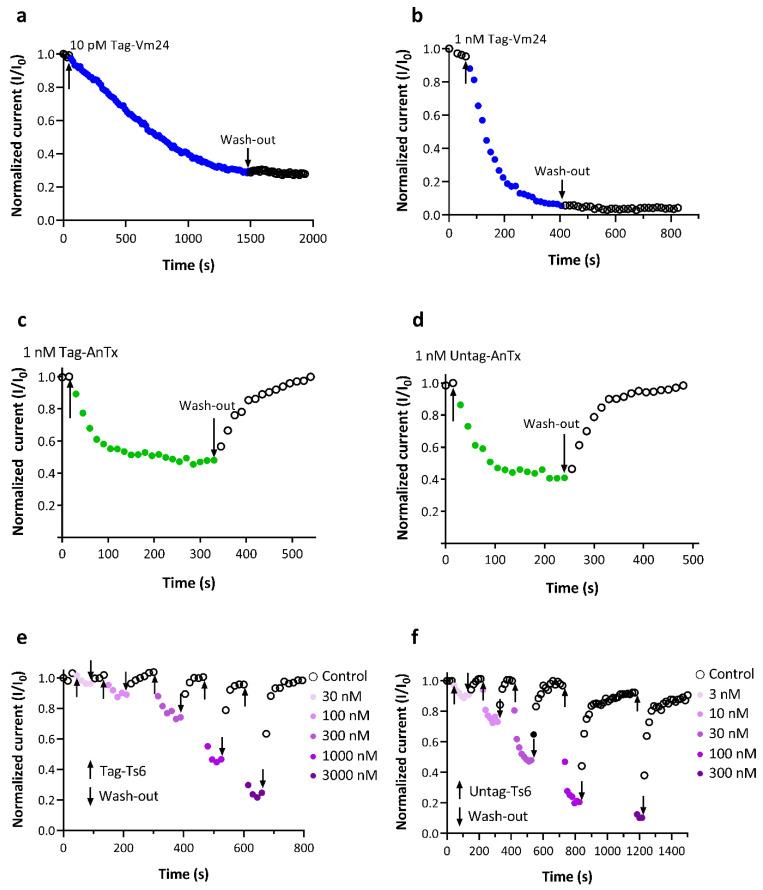
Blocking kinetics of recombinant Vm24, AnTx, and Ts6 on hKv1.3. Whole-cell currents through hKv1.3 were evoked from activated human peripheral T lymphocytes by depolarization to +50 mV from a holding potential −120 mV for 15 ms duration. Test pulses were applied every 15 s. Time courses of the development and recovery of the hKv1.3 current inhibition are shown for each peptide. Normalized peak K^+^ currents (I/I_0_) were determined and plotted as a function of time. Down and up arrows indicate the start and the end of perfusion with the recombinant peptide containing solution or toxin-free solution, respectively. Data points in blue, green, or purple shades represent the addition application of recombinant Tag-Vm24 (panel (**a**,**b**)), AnTx (panel (**c**,**d**)), and Ts6 (panel (**e**,**f**)), respectively, in their tagged or untagged version, as indicated.

**Figure 6 jof-08-01215-f006:**
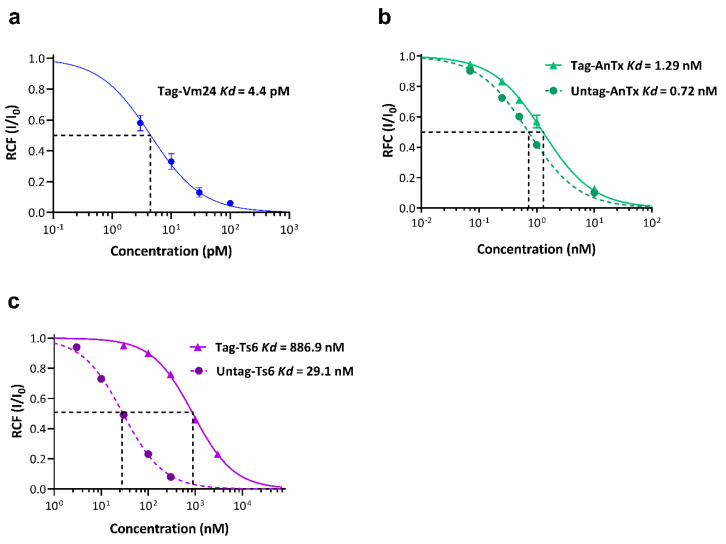
Concentration-dependent block of hKv1.3 by recombinant Vm24, AnTx, and Ts6. A Hill equation (see Materials and Methods for details) was fitted to the averaged remaining current fraction (RCF) values calculated at different toxin concentrations as indicated. The best fit resulted in the Kd values shown at the top of each graph. Hill coefficients were H = 0.92 for Tag-Vm24 (panel (**a**)), H = 0.91 for Tag-AnTx, H = 0.95 for Untag-AnTx (panel (**b**)), H = 1.06 for Tag-Ts6, and H = 1.03 for Untag-Ts6 (panel (**c**)). Solid lines represent tagged peptide, dashed lines represent untagged peptides. Error bar represents SEM and n = 3–5.

**Figure 7 jof-08-01215-f007:**
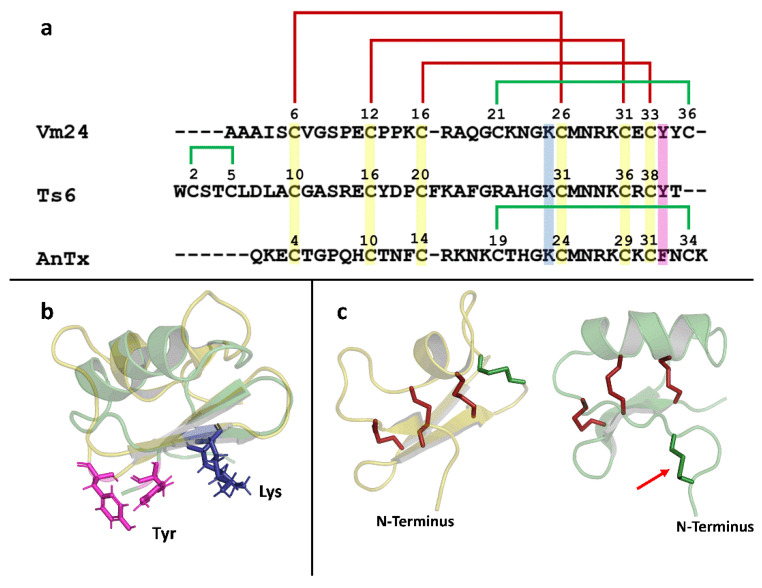
Structural analysis of Vm24, Ts6, and AnTx. (**a**) Sequence alignment of Vm24, Ts6, and AnTx. Conserved Cys between the three peptides are highlighted in yellow; the functional dyad is highlighted in blue and pink. Conserved disulfide bridges are shown in red; the extra disulfide bridges are shown in green. (**b**) Structural alignment between Vm24 and Ts6. Vm24 structure is shown in yellow (PDB:2K9O); Ts6 structure is shown in green (PDB:1C56); functional dyads are shown in the structural alignment as sticks, lysine (blue) and tyrosine (pink). (**c**) Disulfide pattern of Vm24 and Ts6. Conserved disulfide bridges are shown in red sticks; the other disulfide bridges are shown in green. The red arrow points to the disulfide bridge found in the N-terminus of Ts6.

**Table 1 jof-08-01215-t001:** Initial characterization of the recombinant Vm24, AnTx, and Ts6.

Peptide	RT (min)	CH_3_CN %	TMM (Da)	EMM (Da)	Activity on hKv1.3
Tag-Vm24	31.8	23.4	5656.549	5656.500	Positive
Tag-AnTx	32.7	21.2	6025.615	6025.644	Positive
Tag-Ts6	36.8	25.9	6189.546	6189.490	Positive

RT: retention time in RP-HPLC chromatogram; CH_3_CN %: acetonitrile % of elution in RP-HPLC chromatogram; TMM: theoretical molecular mass; EMM: experimental molecular mass.

**Table 2 jof-08-01215-t002:** Comparison of tagged and untagged peptide Kd values on hKv1.3.

Peptide	Tagged	Untagged	Native	
AnTx	1.29 nM	0.73 nM	0.72 nM	[25]
Vm24	4.4 pM	NC	2.9 pM	[13]
Ts6	886.9 nM	29.1 nM	0.55 nM	[26]

NC: Not calculated.

## Data Availability

The raw data supporting the conclusions of this article will be made available by the authors, without undue reservation.

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
