# Peer review of "Recombinant Expression in Pichia pastoris System of Three Potent Kv1.3 Channel Blockers: Vm24, Anuroctoxin, and Ts6"

_jof, 2022, doi:10.3390/jof8111215_

Round 1

Reviewer 1 Report

The authors report on heterologous expression of three Kv1.3 channel blockers in Pichia pastoris. The three peptides were expressed with an N-terminal His-tag and secreted using the α-mating factor of S. cerevisiae. Purified tagged and untagged peptides were evaluated for their biological activity. Kd values of tagged Vm24 and untagged AnTx were similar to those reported for the native ones.  

The manuscript is well written and the results are clearly presented. The findings are of interest for the readers in this field. I would suggest to accept this manuscript after some minor corrections.  

Minor remarks:

-          The authors should mention that Pichia pastoris has been recently reclassified as Komagataella phaffii.

-          Fig. 2: Each lane should be labelled;

-          Fig. 3: SDS-PAGE analysis of purified peptides should be added

-          line 298-299: The authors should make clearer whether the expression yields were calculated for the tagged or untagged peptides and before or after purification

-          line 301-302: expression yields for all peptides should be compared to cell-free and heterologous expression of these peptides as reported in literature

-          line 310: SDS-PAGE analysis of untagged peptides after purification should be provided in SI

Author Response

First of all, we would like to thank Reviewer 1 for the time and effort devoted to reviewing our manuscript, and we thank the critical comments and suggestions for the restructuring of the manuscript. Please find our itemized answers below:

  • The authors should mention that Pichia pastoris has been recently reclassified as Komagataella phaffii.
    • Answer: Thank you very much for the correction, we included this information into the manuscript in Line 65
  • 2: Each lane should be labelled;
    • Answer: We have corrected Fig 2. as per the suggestion
  • 3: SDS-PAGE analysis of purified peptides should be added
    • Answer: Thank you very much for the suggestion, we included this data as panel b in Fig 3. (Fig 3b)
  • The authors should make clearer whether the expression yields were calculated for the tagged or untagged peptides and before or after purification
    • We put the requested information into the manuscript (page 8) to read as: “The yields after RP-HPLC purification of tagged Vm24, Ts6, and AnTx..:”
  • Expression yields for all peptides should be compared to cell-free and heterologous expression of these peptides as reported in literature
    • Answer: Thank you very much for the suggestion, we have included a detailed analysis of this issue on page 8 of the revised manuscript (middle of the page, highlighted)
  • SDS-PAGE analysis of untagged peptides after purification should be provided in SI
    • Answer: We have a new figure, Fig. S2 in the supplemental material

Reviewer 2 Report

General Comments

This paper is dealing with recombinant expression in Pichia pastoris system of three 2 potent Kv1.3 channel blockers: Vm24, Anuroctoxin, and Ts6. Authors expressed these peptides in P. pastoris and prove their biological activity as Kv 1.3 channel blockers by patch clamp. This work is well structured. Nevertheless, some sections of the paper can be improved and better displayed. Sime typo errors should be correct. For instance, in line 316, change “normlized” by “normalized”.

Specific Comments

In Results and Discussion section, authors declares that soluble expression of the peptide 222 was not achieved in the cytoplasm, and periplasmic expression yielded only a very low 223 amount of the toxin. Can you suggest some explanation for this? In addition, authors sed different protease sequence sires for removing His-tag of the expressed toxins. Why did you used different sites? In Figure 2, three SDS-PAGE are displayed. However, I think it is needed to clarify that each lane in the images matches to different clones. Authors mention that they carried out SDS PAGE for HPLC fractions to prove peptide purification. I think they should display SDS PAGE images of these results. In addition, ESI QTOF MS spectra for the different peptides studied should be added with their respective intensities (a.u.) and with m/z values in separated figures, and not only the monoisotopic mass of the peptides. In addition, it was confirmed presence of stable disulfide linkages in the peptides? Auhors shoul include sequences and 3D structure of native peptides in supplementary information.

Authors express that His-tag can “sometimes affect protein structure and function”. I think this effect should be explained in depth supported with structural data or even additional work. e.g. docking between peptide and Kv1.3 channel to understand which amino acids are interacting in this blocker toxin-receptor interaction. I think authors should include data about IC50 of toxins to compare dose response experiments and not only kd constants. These toxin peptides are previously studied in their biological and structural properties, and their action mechanism action is kwon. I think authors should discuss about different structure of peptides and their effects on dose responses in terms of structural features. This is relevant specially for Ts6 toxin. If well authors compare Ts6 and Vm2 peptides (See Figure 7) to explain differences in kd values and biological activity of toxins with His tag, I think authors should align structurally Ts6 peptides with and without His-tag. Why are so marked kd values among His-tagged, non-his-tagged and native Ts6 peptide? Some additional explanation for this? Authors argue that type of protein channel or cell used could explain this effect. However, values are in quite different magnitude orders.

Reviewer 3 Report

In this manuscript, the authors showed  developed an efficient system for the production of three potent Kv1.3 channel blockers from different scorpion venoms: Vm24, AnTx, and Ts6. The approach described here provides a method that can be optimized for toxin production to develop more selective and effective Kv1.3 blockers with therapeutic potential. It is a nice work with high quality of presentation. I just have several minor comments as follow:

Minor comments:

1.       Please describe most of  the important chemicals used in this work in the materials and method part, especially the catalog number, so that others could repeat it if necessary.

2.       Just curious, why did the authors add the His tag at the N terminus, but not C terminus.

3.       Figure 2, why the SDS-Page of Antx is cleaner that the other two? Also, the staining may not so sensitive, if possible, I suggest the authors do the western blot by using the HIS tag antibody.

4.       It seems that untagged peptides have the lower Kd values, which means the better application. Just curious, how about using tandem-repeat strategy?

Author Response

First of all, we would like to thank Reviewer 3 for the time and effort devoted to reviewing our manuscript, and we thank the critical comments and suggestions for the restructuring of the manuscript. Please find our itemized answers below:

  • Please describe most of the important chemicals used in this work in the materials and method part, especially the catalog number, so that others could repeat it if necessary.
    • We have completed the missing data as per the suggestion.

  • Just curious, why did the authors add the His tag at the N terminus, but not C terminus.
    • The binding mechanism of these type of toxins to the potassium channel significantly involves the C-terminal region of the peptide. Therefore, the His tag may interfere with the proper interaction of the peptide with the channel. This justifies the conjugation of the tag to the N-terminus
  • Figure 2, why the SDS-Page of Antx is cleaner that the other two? Also, the staining may not so sensitive, if possible, I suggest the authors do the western blot by using the HIS tag antibody.
    • Answer: For Vm24 and Ts6 gels, 1 ml of sample was precipitated from the supernatant after culturing, and the pellet was resuspended and loaded onto the gel. For AnTx, a sample extracted from the unprecipitated supernatant (30 µL) was used. In the precipitated samples, the other yeast proteins are also concentrated and can be easily observed in the gel. In contrast, in the case of AnTx, only the recombinant protein is observed due to its high expression, and the other proteins are too dilute to be noticed.

The Reviewer is right, we could use the anti-HIS antibody to further characterize the products. However, after all the experiments performed, including MS and patch-clamp, we can conclude that the peptides obtained are the ones that were expected. As for the sensitivity, we rather used the lower sensitivity Coomassie stating as an indicator of a culture with significant peptide content, i.e., a signal with Coomassie staining indicated that the culture originated from a colony with high expression yield. A high sensitivity readout, such as western blot, may not distinguish sufficiently between high-yield and low-yield colonies and therefore western blot was not used.

  • It seems that untagged peptides have the lower Kd values, which means the better application. Just curious, how about using tandem-repeat strategy?
    • Answer: we think this would have the same implications as using a Tag in the C-terminal region of the toxin. With the tandem-repeats, we think that only the molecule that has the free C-terminus would have the appropriate structure to block the channel.